# An Efficient GNSS Coordinate Classification Strategy with an Adaptive KNN Algorithm for Epidemic Management

**Jong-Shin Chen \*** and **Chun-Ming Kuo**

Department of Information and Communication Engineering, Chaoyang University of Technology, Taichung 413310, Taiwan
* Correspondence: jschen26@cyut.edu.tw

**Abstract:** In times of widespread epidemics, numerous individuals are at risk of contracting viruses, such as COVID-19, monkeypox, and pneumonia, leading to a ripple effect of impacts on others. Consequently, the Centers for Disease Control (CDC) typically devises strategies to manage the situation by monitoring and tracing the infected individuals and their areas. For convenience, "targets" and "areas" represent the following individuals and areas. A global navigation satellite system (GNSS) can assist in evaluating the located areas of the targets with pointing-in-polygon (PIP) related technology. When there are many targets and areas, relying solely on PIP technology for classification from targets to areas could be more efficient. The classification technique of k-nearest neighbors (KNN) classification is widely utilized across various domains, offering reliable classification accuracy. However, KNN classification requires a certain quantity of targets with areas (training dataset) for execution, and the size of the training dataset and classification time often exhibit an exponential relationship. This study presents a strategy for applying KNN technology to classify targets into areas. Additionally, within the strategy, we propose an adaptive KNN algorithm to enhance the efficiency of the classification procedure.

**Keywords:** KNN; GNSS; pointing-in-polygon; machine learning; epidemic management

**MSC:** 68T07

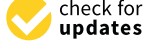



## 1. Introduction

Infectious diseases are diseases caused by microorganisms [1]. Many infectious diseases, such as COVID-19, monkeypox, chickenpox, and influenza, are highly contagious and seriously affect human health, economic activities, education, sports, and leisure. Restricting, tracing, and isolating the movement of people (targets) during an epidemic is an effective way to slow its spread [2,3]. Knowing the areas of targets can help allocate medical treatment or protection-related materials or personnel. GNSS technology is also very mature and can accurately provide the current geographical coordinates of targets. These coordinates can be converted into the areas where the targets are located. In this paper, we will propose a coordinate classification strategy, which uses the GNSS coordinates of the current targets and KNN technology to classify (predict) the areas where the targets are located.

A GNSS refers to a satellite navigation system providing worldwide coverage. It enables tiny electronic receivers to determine their longitude and latitude coordinates. In the context of international civil aviation standards, there are currently two primary constellations recognized: the U.S. global positioning system (GPS) and the global navigation satellite system (GLONASS) [4,5]. A GNSS allows a tiny electronic receiver to determine its current coordinates, including its longitude, latitude, and altitude, to an accuracy of a few centimeters to a few meters by using time signals transmitted by satellite radio along the line of sight [4,5]. Continuing, much research is devoted to more accurate GNSS

positioning [6–8]. In today's environment, GNSS receivers are a widely used piece of equipment. They widely exist in people's personal belongings, such as mobile phones, tablet computers, and watches. Therefore, obtaining the current GNSS coordinates of targets and extending them to the current areas of targets is available. Due to their ability to furnish accurate target location data, GNSSs have found extensive application across a range of remote sensing endeavors [9–15], including but not limited to epidemic monitoring and geographic information systems.

The GNSS can provide a two-dimensional coordinate of the longitude and latitude of a target, which can be regarded as a point in a two-dimensional plane in geography. An area in the range of geography can be represented by a polygon composed of multiple points. Determining whether a point lies inside a polygon is a geometric problem known as the PIP calculation. These methods can generally be distinguished by two types: the ray casting and the winding number. These methods have their own advantages and disadvantages according to the spatiality of the polygon, but their time complexity is generally O(n), where n is the number of vertices of the polygon. These methods can be developed into different procedures to calculate points that are inside or not inside the polygon. We simply call these procedures the PIP_EPs (PIP evaluation procedures). Because the PIP_EPs continue to have promising applications in many fields, related technologies continue to be developed [16–23].

In order to know the spatial relations of the located polygons of the candidate points, a PIP_EP is necessary. Assuming there is only one specified polygon and many candidate points, planning an enumeration method with an existing PIP_EP is also a solution. Allowing each candidate point to enter this PIP_EP once achieves the result, but such a method could be more efficient, especially when the final internal points are only a tiny part of all points. In [22], the author planned a similar enumeration method with some candidate points excepted. In [23,24], the methods can obtain most points inside this polygon without entering the PIP_EP. When there are many candidate points and polygons simultaneously, it is very complicated to accurately calculate their spatial relationships. Moreover, in the application of epidemic management, the number of candidate points and polygons is large and constantly changing. Prediction in machine learning provides a direction to try.

KNN classification is an efficient solution to approximation, which is widely used in various fields [25–30]. KNN classification has the remarkable property that, under very mild conditions, the error rate of a KNN algorithm tends to be Bayes optimal as the sample size is towards infinity [25]. If establishing a model with some training datasets is troublesome for any data analysis application, a KNN algorithm will likely provide the best solution [26]. In addition, KNN is also widely used in the fields of data mining [27] and artificial intelligence (AI) [28,29]. To provide classification, a training dataset as samples is needed. Each data point in this dataset contains multiple features and has its class. The experimental results in [30] have shown that when the number of samples is large, and the dimension of data points is low, the classification accuracy presented by KNN classification will be higher. For epidemic management, the targets' locations meet the KNN classification feature. In [31], the author proposed a KNN-based GNSS coordinate classification method. The time complexity of this classification is O($n_{TD}^2$), where $n_{TD}$ is the size of this training dataset. From previous experience, we know that weighting KNN technology can improve classification accuracy in many applications, especially when the training dataset is insufficient. As the training dataset increases, the accuracy of KNN classification will also increase. Unfortunately, the classification time will show an exponential relationship with the size of the training dataset, and the classification accuracy of KNN and the classification time of KNN are in a trade-off relationship. This result shows that there are several improvements. The first is to improve the accuracy when the number of samples is insufficient. The second is to reduce the time complexity of the KNN classification.

The classification of geographic areas based on the latitude and longitude coordinates provided by GNSS is a novel research topic [23,24]. KNN represents a conceptual approach

to classification (or prediction) that extends into new research areas, as seen in recent studies [32–34], which apply KNN in various domains. Our research aligns with this trend. In our survey, we are the first to introduce the concept of KNN for classifying many latitude and longitude coordinates provided by GNSS into numerous geographical areas. Additionally, we incorporate the idea of weighting KNN [30,35] into this study to increase the classification accuracy when the number of samples is small. Regarding classification accuracy, the concept of both KNN and weighting KNN are beneficial in this field. However, the significantly large number of classes and the corresponding expansion of training data points in this study have impacted the efficiency of classification time. Therefore, we also need to propose an adaptive KNN method for classification time.

Research on KNN classification often focuses on the issue of adaptability. In our survey, studies [36–44] can be roughly categorized into two types. The first type involves adapting the k value based on the precise knowledge of the samples [36–40], where *k* is a specific number that uses *k* data points of the training dataset to classify a test point. Such studies tend to lean towards theoretical exploration and technical extension, such as adjusting the k value based on the underlying distribution of samples to achieve an optimal minimax rate [36] or a desirable bias and variance tradeoff [38]. The second type extends adaptive KNN classification methods based on problems arising in specific applications [41–43]. For example, the method mentioned in [41] is designed for predicting diabetes, and, in [43], the technique aims to improve the retrieval rate of leaf datasets. This study focuses on using a test point as the center, employing a distance r as the range, and calculating the data points within the range through a single search of the training dataset. If the number of data points within this range is less than k, the distance r will adaptively adjust until the number of these data points is greater than or equal to k. These points become candidates for the *k* points. Another approach is to consider all of the training points as candidates. Our method performs better during classification time by reducing the number of candidates to a very low level.

Machine learning plays a crucial role in artificial intelligence applications [44,45], with deep learning achieving excellent results in various fields such as computer vision, speech recognition, natural language processing, audio recognition, and bioinformatics. The deep learning algorithm utilizes artificial neural networks as its framework for representation learning. When applied to classification tasks with a sufficiently large dataset, frameworks predominantly based on neural networks often provide reliable classification accuracy. However, neural networks also have limitations. For instance, training a neural network requires substantial computational resources, and both the training process and classification results often lack interpretability, which implies challenges in assessing the time required for decision making. In the context of the proposed paper, which involves making efficient decisions for many data points, a neural network for classification may not meet these criteria.

The KNN algorithm is one of the simplest among all machine learning algorithms, suggesting that, compared with other classification methods, KNN classification is efficient in terms of time. Additionally, the error rate of a KNN algorithm tends to be Bayes optimal as the sample size approaches infinity [25], which implies that the time for classification, the size of the training set, and the accuracy of classification should be interpretable, which is a primary reason for choosing this classification method. Another example is fuzzy. The application of fuzzy techniques to original data or final results with probabilistic or possibilistic characteristics is representative. However, in this study, the original data consist of geographical points (domain) and geographical areas (codomain), where a geographical area is a set of geographical points. Moreover, the relation between the domain and codomain is clear and does not possess probabilistic or possibilistic characteristics.

Based on the above discussion, we propose an effective GNSS coordinate classification strategy using an adaptive KNN algorithm for epidemic management. This strategy contains two phases. The first phase starts when many areas need attention due to the epidemic. In this phase, when the coordinates of targets enter, these coordinates are converted into

corresponding areas (classes). When enough targets have their coordinates and classes, these data become the training dataset in the second stage. When there is a sufficient training dataset, the second phase starts. This phase will classify the targets generated next. For classification, we introduce the weighting KNN technology. In addition, we also plan an adaptive algorithm in order to improve the classification time of weighting KNN.

## 2. Related Work

A GNSS provides worldwide coverage and currently consists of two prominent constellations: the U.S. global positioning system (GPS) and the global navigation satellite system (GLONASS) [4,5]. Numerous studies have employed various techniques to enhance the precision of GNSS positioning [6–8]. Due to their ability to provide accurate target location data, GNSS systems have found extensive applications in remote sensing endeavors, including prevention and control efforts [9–15]. For instance, in [9], methods are proposed to track detailed movement patterns of individuals using GPS data, which can be integrated into a geographic information system (GIS) to identify high-endemic areas and high-risk groups, facilitating resource allocation decisions [10]. In [12], the study focuses on using GNSS satellites for environmental monitoring, while [13] highlights using smartphones to monitor COVID-19 patients and confirmed cases. Additionally, Ref. [14] emphasizes the importance of mapping flood dynamics and extent in various contexts, including predicting the spread of infectious diseases, with GNSS technology playing a valuable role. Finally, Ref. [15] presents an analysis of COVID-19 trackers in India.

Determining whether a target is inside or outside a specific area can be framed as a PIP problem. Many works in the literature have discussed related issues in the past [16–21]. In [16], the study presented a detailed discussion of the PIP problem for arbitrary polygons. Since no single algorithm is the best in all categories, the study compared the capabilities of different algorithms. In [17], the authors discussed the variables examined, including various polygon types, memory usage, and preprocessing costs. In [18], the authors outlined a fast and efficient method for determining whether a coordinate point lies within a closed region or polygon defined by any number of coordinate points. In [19], the study proposed an efficient polygon clipping algorithm using a parametric representation of polygon edges and point clipping to find required intersection points with clip window boundaries. In [20], the study presented an extension of a winding number and PIP algorithm. Study [21] provided a provably correct implementation of a PIP method based on the computation of the winding number. We can follow the methods in [16–21] to design different PIP_EPs. These PIP_EPs share a common characteristic in that they can only determine whether a point is inside a specified polygon one at a time. When calculating points within a designated polygon from a large set of candidate points, employing an enumeration approach with one of these PIP_EPs to process them one by one is a solving method. However, this method is inefficient. In [22–24], these methods used preprocessing to reduce the number of candidate points and improve efficiency.

In [22], a rectangle covering the targeted area was proposed, and only candidate points inside this rectangle needed to enter the PIP_EP. This method is more effective than the enumeration approach but still has limitations. Study [24] presented a method that significantly improves efficiency compared with [23]. It involves planning subareas covering the target area and calculating the spatial relationships between them and polygons, considering inner, outer, and intersected relationships. This method only allows candidate points in intersected subareas to enter the PIP_EP.

Moreover, it considers the computer's computing power and data access capabilities for greater efficiency. However, this method has two problems for improvement, including the overlapping among subareas and the subarea size. Study [24] effectively addressed these two shortcomings.

KNN classification exhibits several remarkable properties [25–31]. The classification is available in various fields, including data mining algorithms [27] and artificial intelligence (AI) [28,29]. In [31], the author proposes a KNN-based GNSS coordinate method, which

includes a PIP_EP and KNN classification. The PIP_EP is based on a casting type that acquires the training dataset. In the KNN classification, three steps are involved for a test data point as follows:

1. It evaluates the Euclidean distance between the sample and the test point, with a time complexity of $O(n_{TD})$, where $n_{TD}$ is the dataset size.
2. It sorts the training dataset based on Euclidean distances with an $O(n_{TD}^2)$ time complexity.
3. It uses the majority classification rule to predict the class of the test point, with a time complexity of $O(k)$, where $k$ is the number of neighbors in the KNN classification.

Overall, the time complexity of this KNN classification method is $O(n_{TD}^2)$.

The KNN classification technique is widely applied in various fields, and the characteristics of the raw data, such as the number of classes, average size of training data points per class, and dimensions, often impact the classification performance. In [30], the authors conducted experiments on 15 datasets with different characteristics of raw data. These experimental results can be used for cross-referencing with our experiments to assess the suitability of the KNN classification technique for application in this context. Table 1 provides the information about the experimental results with results. These datasets include between 2 and 26 classes, with 5 datasets including 10 or more classes. In the experiment, the average classification accuracy of KNN is between 4.71% and 74.92, and that of weighting KNN is between 5.38% and 73.61%. It is worth mentioning that in 13 of the 15 datasets, weighting KNN improved the accuracy, but in 2 of them it did not. The above data provide a reference for evaluating KNN technology to classify geographical coordinates into geographical areas.

**Table 1.** The KNN classification accuracy of 15 datasets with different data attributes [30].

| Datasets | Classes | Size of Training Datasets | Average Size of Training Data Points per Class | Dimensions | Accuracy | |
|---|---|---|---|---|---|---|
| | | | | | KNN | Weighting KNN |
| OCCUDS | 10 | 645 | 64 | 101 | 13.02% | 14.41% |
| Chess | 2 | 335 | 167 | 36 | 69.00% | 72.90% |
| CNAE | 9 | 376 | 42 | 856 | 13.65% | 14.51% |
| German | 2 | 137 | 69 | 20 | 66.20% | 89.70% |
| Ionosphere | 2 | 45 | 23 | 34 | 83.60% | 87.50% |
| Isolet | 2 | 168 | 84 | 617 | 68.10% | 74.50% |
| Letter | 26 | 6471 | 249 | 16 | 4.71% | 5.38% |
| Segment | 7 | 820 | 117 | 19 | 19.43% | 20.03% |
| USPS | 10 | 3109 | 311 | 256 | 15.32% | 16.71% |
| Vehicle | 4 | 276 | 69 | 18 | 31.05% | 36.33% |
| Waveform | 3 | 684 | 228 | 21 | 49.86% | 58.39% |
| Yeast | 10 | 510 | 51 | 1470 | 48.99% | 42.12% |
| Arcene | 2 | 20 | 10 | 10,000 | 75.00% | 74.10% |
| Carcinom | 11 | 53 | 5 | 9182 | 20.61% | 26.57% |
| CLLAUB | 3 | 33 | 11 | 11,340 | 74.92% | 73.61% |

Research on KNN classification often centers around the issue of adaptability [36–43], broadly including the *k* values' adjustment type [36–40] and application-oriented type [41–43]. The *k* values' adjustment type often provides precise knowledge of the samples, such as the probability density function (pdf) of the samples. In [36], a sliced nearest neighbor method is proposed that requires the pdf of the samples to achieve optimal minmax rate. In [37], the pdf function can be estimated by unclassified samples and then the optimal *k* can be acquired. In [38], the *k* value is selected according to the training samples to achieve

a desirable bias and variance tradeoff. Moreover, in [39], different *k* values are selected for different test samples without any knowledge of the underlying distribution. Continuously, study [40] also pays attention to this issue. For the application-oriented type, in [41], a distance adaptive-KNN method is proposed for predicting diabetes. In [42], a wrapper-based binary improved grey wolf optimizer approach is developed for categorizing Parkinson's disease with an optimal set of features using adaptive KNN. In [43], adaptive KNN for an optimization method is proposed to improve the retrieval rate of leaf datasets. The motivations driving these methods in this category vary, making it challenging to cross-compare the resulting approaches.

### 3. System Model

In this system model, each target needs a mobile device with a tiny GNSS receiver. It also needs a server with a data storage device. The server provides related calculations and data storage required for the location management of targets. The target's mobile device will regularly receive GNSS signals, convert them into coordinate data, and then send these data to the server. A target with latitude and longitude coordinates is a geographic point. For a geographical point (target) $g$, $(g_x, g_y)$ represents its two-dimensional coordinate. The considered areas are in the form of polygon data types. A polygon $P$ is composed of $n$ edges with $n$ geographical points. For convenience, we use $P = \{p_0, p_1, \ldots, p_{n-1}, p_n\}$ to define a polygon $P$ with $n$ points, where $p_0 = p_n$ and the straight line segment from point $p_i$ to $p_{i+1}$ is the edge for $i = 0, 1, \ldots, n - 1$.

Moreover, we also assume $P_A = \{P_0, P_1, \ldots, P_{m-1}\}$ is a set of $m$ polygons in all areas. As shown in Figure 1, the system model includes a platform for finishing the task of a two-phase procedure, i.e., the positioning and classification that a computer server could implement. The considered areas are generated and transformed into the data with polygon type and the system starts. The positioning phase performs the following.

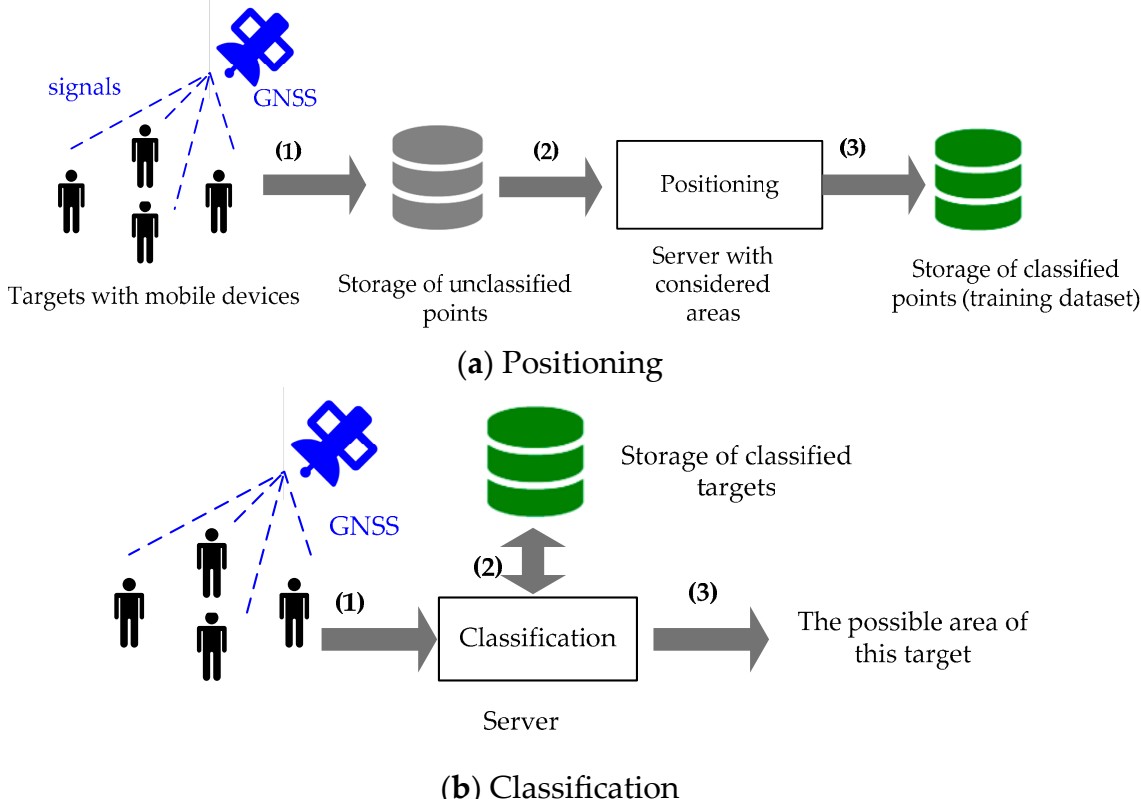

**(a)** Positioning

**(b)** Classification

**Figure 1.** System model. It includes a platform for completing a two-phase procedure, i.e., positioning and classification.

1. The data sent from the mobile devices of targets will be stored as unclassified geographic points.
2. The server will individually take out a point from the unclassified points and position which polygon this point is inside.
3. The server will store these geographical points with their classes.

The task of this phase is to collect and position sufficient training datasets for the classification phase task.

The classification procedure entails selecting candidate samples from the training dataset based on their proximity to the target's coordinates and evaluating the target's assigned area (class). The classification phase performs the following.

1. The real-time data sent from targets' mobile devices arrive to the server.
2. The server extracts candidate points from the storage of classified targets.
3. The server will classify the targets into their located areas according to the candidate points.

## 4. Proposed Algorithms

The proposed algorithms include a PIP positioning algorithm and a KNN classification algorithm.

### 4.1. PIP Positioning Algorithm

Given a point $g$ and a polygon set $P_A = \{P_0, P_1, \ldots, P_{m-1}\}$, where point $g$ is inside a polygon $P$ of set $P_A$, the goal of positioning is to assess this polygon $P$ from set $P_A$. We use a positioning technique with ray-casting technology to achieve this goal. This positioning method determines the number of intersections between the ray of point $g$ and the candidate polygon.

Figure 2 illustrates a polygon with points and their rays. This polygon $P$ consists of six points labeled as $p_0, p_1, \ldots,$ and $p_6$, where $p_0$ is equal to $p_6$. The line segment connecting point $p_i$ to $p_{i+1}$ forms the edge for $i = 0, 1, \ldots,$ and 5. There are three geographic points $g_0, g_1,$ and $g_2$ with the corresponding rays $ray(g_0), ray(g_1),$ and $ray(g_2)$. The number of intersections between $ray(g_0)$ and edges of $P$ is 1 (odd number) that $g_0$ is the inner point of $P$. The numbers of intersections of $ray(g_1)$ and edges of $P$ and $ray(g_1)$ and edges of $P$, respectively, are 2 and 0 that both numbers are even. It means that point $g_1$ and point $g_2$ both are the outer points of $P$.

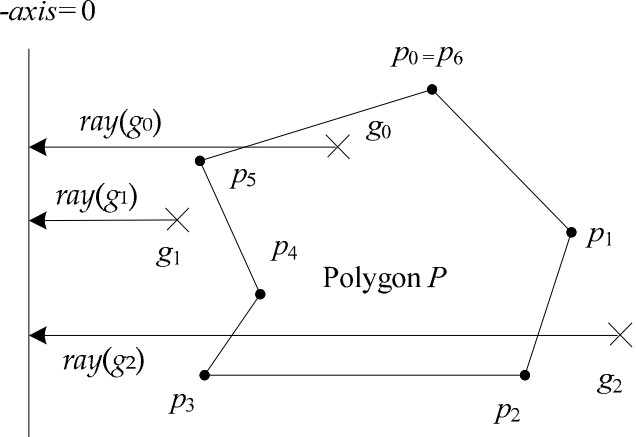

**Figure 2.** Example of a polygon with points and their rays.

In order to count these intersections, a procedure for evaluating the intersection of two lines is required [46]. Consider two lines: line 1 and line 2. Line 1 intersects points $g$a and $g$b, and line 2 intersects points $g$c and $g$d. The intersection between the lines is according to (1), where if $\alpha$ is 0, there is no intersection; otherwise, there is an intersection.

The intersection of line 1 is according to (2), where if κ1 is between 0 and 1, then the intersection is between points $g$a and $g$b. The intersection of line 2 is according to (3), where if κ2 is between 0 and 1, the intersection is between points $g$c and $g$d. Correspondingly, the condition for the intersection of two line segments from points $g$a to $g$b and points $g$c to $g$d is that both κ1 and κ2 are between 0 and 1. Algorithm 1 formally expresses the procedure to evaluate the intersection of two line segments (SegSegInt). Given two line segments from points $g$a to $g$b and points $g$c to $g$d, according to the procedure of SegSegInt, it can result in 0 or 1, indicating that the two segments are intersected or not intersected.

$$\alpha = (g a_x - g b_x) \times (g c_y - g d_y) - (g a_y - g b_y) \times (g c_x - g d_x) \tag{1}$$

$$\kappa 1 = ((g a_x - g c_x) \times (g c_y - g d_y) - (g a_y - g c_y) \times (g c_x - g d_x))/\alpha \tag{2}$$

$$\kappa 2 = ((g a_x - g b_x) \times (g a_y - g c_y) - (g a_y - g b_y) \times (g a_x - g b_x))/\alpha \tag{3}$$

---

**Algorithm 1:** SegSegInt (point $g$a, point $g$b, point $g$c, point $g$d).

---

**Input:** $g$a, $g$b, $g$c, $g$d are points that form a segment from $g$a to $g$b and a segment from $g$c to $g$d
**Output:** The result is 1 or 0, indicating which two line segments intersect or do not.
**Method://**an algorithm for evaluating the intersection of two segments
1.    *result* := 0;
2.    $\alpha := (g a_x - g b_x) \times (g c_y - g d_y) - (g a_y - g b_y) \times (g c_x - g d_x)$;
3.    **if** $\alpha = 0$ **then**
4.        *result* := 0;
5.    **else**
6.        $\kappa 1 := ((g a_x - g c_x) \times (g c_y - g d_y) - (g a_y - g c_y) \times (g c_x - g d_x))/\alpha$;
7.        $\kappa 2 := ((g a_x - g b_x) \times (g a_y - g c_y) - (g a_y - g b_y) \times (g a_x - g b_x))/\alpha$;
8.        **if** $(\kappa 1 \geq 0$ and $\kappa 1 \leq 1)$ and $(\kappa 2 \geq 0$ and $\kappa 2 \leq 1)$
9.        **then**
10.           *result* := 1;
11.        **else**
12.           *result* := 0;
13.        **end if**
14.    **end if**
15.    **output** *result*.

---

In order to calculate the number of intersections of the ray of point $g$ with polygon $P = \{p_0, p_1, \ldots, p_{n-1}, p_n\}$, a line segment from points $g$ to point $g'$ can simplify the ray of point $g$ with coordinate $(g_x, g_y)$, where the y-coordinate of point $g'$ is the same as $g_y$ and the x-coordinate value is 0. For an edge from $p_i$ to $p_{i+1}$ of polygon $P$, where $i = 0, 1, \ldots, n$ and the line segment from points $g$ to $g'$, according to the procedure of SegSegInt, it can acquire this relation of intersection between this edge and the line segment. Expanding the line segment to each edge of polygon $P$ with SegSegInt can achieve the number of intersections between the line segment and the polygon $P$. If the number is odd, point $g$ is inside polygon $P$; otherwise, point $g$ is not inside polygon $P$. Algorithm 2 formally expresses our proposed procedure of the PIP_EP. Given a point $g$ and a polygon $P$, according to the procedure of PIP_EP, it can result in 0 or 1, indicating that point g is inside or not inside polygon $P$.

Since the located polygon of point g is indeed a polygon of the set $P_A$, an enumeration procedure can solve this problem by passing the polygons of set $P_A$ one by one through PIP_EP. The polygon to which the point belongs can be determined. Algorithm 3 formally expresses the procedure of point positioning (PtPos). Given a point $g$ and a polygon set $P_A$, according to the procedure of PtPos, it can result in a number $i$ indicating point $g$ inside polygon $P_i$ of set $P_A$.

---

**Algorithm 2:** PIP_EP (point g, polygon P).

---

Input: $g$ is a point and $P = \{p_0, p_1, \ldots, p_n\}$ is a polygon.
Output: The result is 1 or 0, indicating that $g$ is located in P or not.
Method://an algorithm for positioning a point to a polygon
1.    *count* := 0; *result* := 0;
2.    **for** $i$ :=0 **to** $n - 1$ **do** //each edge of polygon $P$
3.        $g'_x := 0; g'_y := g_y$;
4.        **if** SegSegInt $(p_i, p_{i+1}, g, g') = 1$ **then**
5.            *count* := *count* + 1;
6.        **end if**
7.    **end for**
8.    **if** (*count*%2 = 1) **then** *result* := 1;
9.    **else** *result* := 0;
10.   **end if**
11.   **output** *result*.

---

**Algorithm 3:** PtPos (point $g$, polygon set $P_A$).

---

Input: $g$ is a point and $P_A = \{P_0, P_1, \ldots, P_{m-1}\}$ is a polygon set.
Output: The result is a value $i$ indicating that point $g$ is located in polygon $P_i$, where $0 \le i \le m-1$.
Method: an algorithm for positioning the located polygon of point g.
1.    **for** $i := 0$ **to** $m - 1$ **do**
2.        **if** PtInPy$(g, P_i) = 1$ **then**
3.            break;
4        **end**
5.    **end for**
6.    **output** $i$.

---

*4.2. KNN Classification*

For a point g, the located polygon is one of $P_A = \{P_0, P_1, \ldots, P_{m-1}\}$. Simply, if point g is located in polygon $P_i$, where $0 \le i \le m - 1$, the class of point $g$ is class $i$ and $g_{pc}$ is used to represent the class of point $g$, i.e., $g_{pc} = i$. Through KNN classification, point g can also have a class. For convenience, we use $g_{cc}$ to represent the class of point $g$ in KNN classification. If $g_{cc}$ is equal to $g_{pc}$, the classification of point g is accurate. For KNN classification, the domain is $P_A$ and the training dataset is $T$. KNN classification involves determining the category of a test point by evaluating a specific number of data points from the training dataset. The process relies on assessing the characteristics of these data points and inferring the result accordingly. For convenience, we use '$k$' to represent this specific number. For a point $g$, let $NB$ be a set of its $k$-nearest neighbors. Moreover, $NB$ is a subset of $T$. The KNN classification for a point $g$ is as (4), where $P_i \in P_A$ and $I(-)$ is an indicator function. For each data point g′ in set $NB$, the KNN classification will make statistics on the class $g'_{pc}$ of $g'$ using $I(-)$, find the class $i$ with the largest number, and then assign it to $g_{cc}$, i.e., $g_{cc} = i$.

$$g_{cc} = \arg_i\left(max \sum_{g' \in NB} I(g'_{pc} = i)\right) \tag{4}$$

In the majority rule, the $k$-nearest neighbors of point $g$ are implicitly assumed to have equal weight, regardless of their relative distance to point $g$. It is conceptually better to give different weights to the $k$-nearest neighbors depending on their distance to point $g$, with closer neighbors having greater weight. The Euclidean distance can be applied to give weights. The Euclidean distance of two points $ga$ and $gb$ is as (5), where $(ga_x, ga_y)$ is the coordinate value of point $ga$ and $(gb_x, gb_y)$ is the coordinate value of point $gb$.

$$d(ga, gb) = \sqrt{(ga_x - gb_x)^2 - (ga_y - gb_y)^2} \tag{5}$$

The weighting KNN classification for point g is as follows:

$$g_{cc} = \underset{i}{\arg}(max \sum_{g\prime \in NB} I(g\prime_{pc} = i) \times d(g, g\prime)^{-1}) \tag{6}$$

Algorithm 4 formally presents a procedure of an adaptive KNN classification (AdaptKNN). Given a point g, a numerical value $r$, a training dataset $T$, a polygon set PA, and a $k$ value, according to the procedure of AdaptKNN, it can result in the class $gcc$ of point $g$. The procedure includes four steps. In lines 1 to 9, Step 1 includes a search of $nb$ neighbors from set $T$. Based on a distance $r\prime$, it searches for points within the distance $0.5r\prime$ from point $g$ and stores them in $NB$, where the initial value of $r\prime$ is set as $r$. If the size $nb$ of $NB$ is less than $k$, it enlarges this distance $r\prime$ by $r$ for this search. After this step, $NB(i)$ denotes the neighbor $i$ of point g, where $i = 0, 1, \ldots,$ and $nb - 1$. In lines 10 to 11, Step 2 assigns weight to each neighbor $NB(i)$, denoted as $NB(i)_w$. In lines 13 to 19, Step 3 is a sort of $NB$ that rearranges the elements of NB according to the weight value from high to low. Step 4 includes two phases that classify point $g$, where $NB(i)_{pc}$ is the class of point $NB(i)$. Step 4-1 is an accumulation of weight for the class of the point in $NB$. After Step 4-1, $V(i)$ is the accumulation of weight of class $i$ for the $k$ neighbors. Step 4-2 is a search of class $gcc$ that the accumulation of weight is largest.

---

**Algorithm 4:** AdaptKNN (point $g$, numerical value $r$, training dataset $T$, $P_A$, integer $k$).

---

**Input:** $g$ is a point, $r$ is a numerical value, $T$ is a training dataset, and $k$ is a specific number.
**Output:** the class of point $g$
**Method:**//an algorithm for adaptive KNN classification
**Notation and Initialization:**

.    $m$: the size of polygon set $P_A$
.    $NB$: a set for storing the neighbors of point $g$, where the arrangement of $NB$ is $((NB(0), NB(1),\ldots)$
.    $V$: a numerical list $(V(0), V(1), \ldots, NB(m - 1))$ for a vote. The initial value of $V(i)$ is 0 for $0 \le i \le m - 1$

1.    $r\prime := 0; nb := 0;$
2.    **while** $nb < k$ **do** /*Step 1: a search of $nb$ neighbors*/
3.      $r\prime := r\prime + r; nb := 0;$
4.      **for each** $g\prime$ in $T$ **do**
5.        **if** $g_x \ge (g\prime_x - 0.5r\prime)$ **and** $g_x \le (g\prime_x + 0.5r\prime)$ **and** $g_y \ge (g\prime_y - 0.5r\prime)$ **and** $g_y \le (g\prime_y + 0.5r\prime)$ **then**
6.          $NB(nb) := g\prime; nb := nb + 1;$
7.        **end if**
8.      **end for**
9.    **end while**
10.    **for** $i := 0$ **to** $nb - 1$ **do**/* Step 2: an assign of weight to each neighbor $NB(i)$ */
11.      $NB(i)_w := 1/d(g, NB(i));$
12.    **end for**
13.    **for** $i := 0$ **to** $nb - 2$ **do** /* Step 3: a sort of $NB$ */
14.      **for** $j := i + 1$ **to** $nb - 1$ **do**
15.        **if** $NB(j)_w > NB(j + 1)_w$ **then** //Swapping of $NB(j)$ and $NB(j + 1)$
16.          $g\prime := NB(j); NB(j) := NB(j + 1); NB(j + 1) := g\prime;$
17.        **end if**
18.      **end for**
19.    **end for**
     /* Step 4: a classification for point $g$*/
20.    **for** $i := 0$ **to** $k - 1$ **do**/* Step 4-1: An accumulation of weight for the class $NB(i)_{pc}$ */
21.      $t := NB(i)_{pc}; V(t) := V(t) + NB(i)_w;$
22.    **end for**
23.    $g_{cc} := 0;$/* Step 4-2: A search of class $g_{cc}$ that the accumulation of weight is largest */
24.    **for** $i := 1$ **to** $n_{PA} - 1$ **do**
25.      **if** $V(i) > V(g_{cc})$ **then**
26.        $g_{cc} := i;$
27.      **end if**
28.    **end for**
29.    **output** $g_{cc}$.

---

Assuming that when there is a test point $g$, the classification procedure will achieve $k$ neighbors after evaluating the nearest neighbors process. When $k$ is 3, the data of $k$ neighbors will be recorded in $NB(0)$, $NB(1)$, and $NB(2)$ and assume that their classes are $P_0$, $P_0$, and $P_1$, respectively. The KNN classification will specify that the class of $g$ is $P_0$. The weighting KNN classification will incorporate the Euclidean distances of the test point g and the three neighbors into the evaluation. Assuming that these Euclidean distances are 4, 2, and 1, respectively, then $V(0)$ is 0.75(=1/4 + 1/2) and $V(1)$ is 1 (=1/1). Because $V(1)$ is greater than $V(0)$, the weighting KNN classification will specify that the class of point g is $P_1$.

Algorithm 4 employs the technology of the weighting KNN classification for classifying points into areas. In addition, Step 1 of this algorithm calculates the candidates of $k$ neighbors based on a numerical value $r$. When necessary, the value of r will be adaptively adjusted until the number of candidates is greater than or equal to $k$. So, the candidates of $k$ neighbors in Steps 2, 3, and 4 are $k$ or slightly more than $k$ data points, not the total training dataset. In this way, we improve the classification time.

*4.3. Analysis*

The proposed strategy includes a positioning method and a classification method. In this positioning method, given two line segments, the SegSegInt is used to evaluate the intersection with time complexity O(1). Given a point and a polygon, the PIP_EP can position whether this point is inside or not inside this polygon. The PIP_EP employs the SegSegInt to evaluate the intersection of the ray of this point and each edge of the polygon, where the time complexity is O($n$), where $n$ is the number of edges of this polygon. Given a point and a polygon set, the PtPos can evaluate the located polygon of this point in which the PIP_EP is used for each polygon of this set until ensuring the located polygon of this point. Assuming $n_{max}$ is this polygon's largest edge number of this polygon set and m is the size of the given polygon set, the PtPos can position a point in O($m \times n_{max}$) time. It follows from the above that we can conclude the following property.

**Property 1.** *Given a point g and a polygon set $P_A$ with size m, if point g is inside one of set $P_A$, Algorithm PtPos positions point g in O(m × $n_{max}$) time, where $n_{max}$ is this polygon's largest edge number of this polygon set.*

The classification method includes four steps. Step 1 includes a search for neighbors. This search collects candidates within a fixed range from the training dataset $T$ that can be achieved in O($n_T$) time, where $n_T$ is the size of dataset $T$. Step 2 is a weight assignment of the candidate points that can be achieved in O($nb$), where $nb$ is the number of the candidate points. In step 3, there are types of candidate points that can be achieved in O($nb^2$) time. In step 4, there is a vote according to the classes of $k$ neighbors that can be achieved in O($k$) time. Since $n_T$ is much greater than $nb^2$ and $nb^2$ also is greater than k, we can conclude the following property.

**Property 2.** *Given a point g and a training dataset T with size $n_T$, algorithm AdaptKNN classifies point g in O($n_T$) time.*

**5. Experiment**

The experimental environment consists of the scope of a geographic area and a set of geographic points within the area. The area is a famous city, ranging from 120.6 to 122.9 east longitude and 24.8 to 25.4 north latitude. The administrative division of this area includes 12 districts or 456 villages, which are 2 types of classes: Type-1 and Type-2.

The initial data for the experiment include geographical points and geographical regions within the selected area. These geographical points, totaling 89,209, are represented by check-in places provided by a well-known social networking platform. We can acquire

the necessary check-in places according to the method in [22]. Taiwan's open data platform can provide the geographical regions of these 12 districts [47] or 456 villages [48]. Figure 3 provides the distribution of the data points, in which the enclosed range is the area and the purple color points are the locations of geographical points, for which 10,000 points are randomly selected from the dataset. Figure 4 provides the class distributions of Type-1 and Type-2.

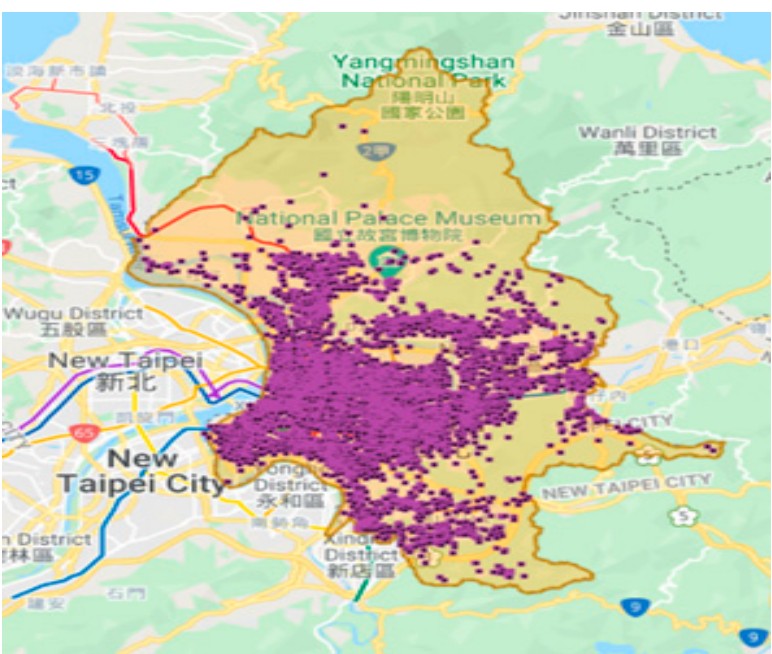

**Figure 3.** The distribution of data points. The non-English terms in this figure are Chinese place names.

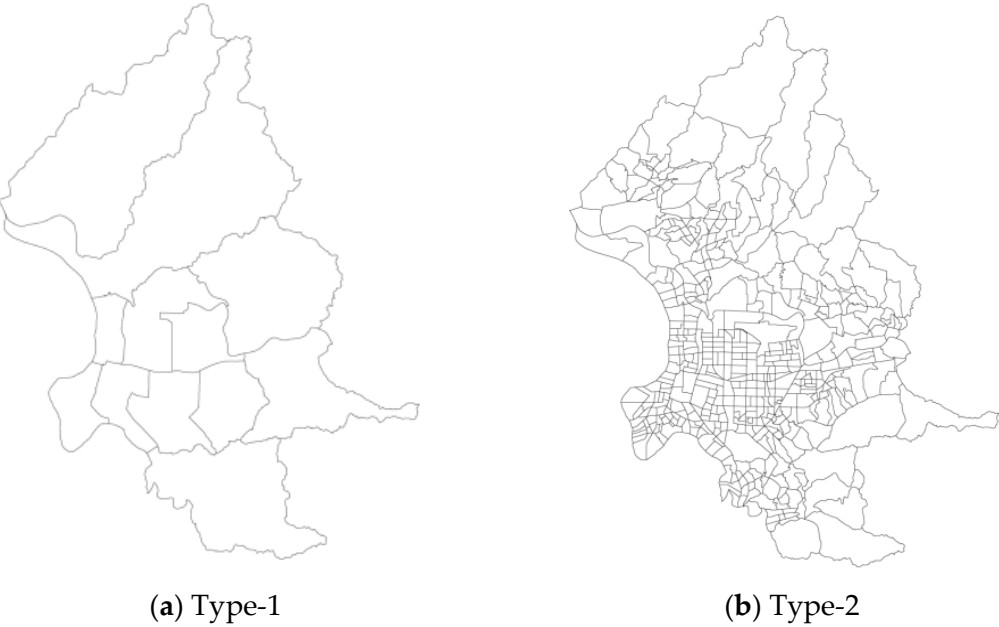

(**a**) Type-1          (**b**) Type-2

**Figure 4.** The class distributions of Type-1 and Type-2. Type-1 has 12 classes and Type-2 has 256 classes.

The other experimental setting for classification includes the $k$ neighbors and the *size*, i.e., the average number of the training data points per class. For each type, the training dataset contains $n \times size$ points, where $n$ is the number of classes and *size* as 2, 4, 8, 16,

32, 64, and 128. Moreover, the values of $k$ include 1, 3, 5, 7, 9, and 11. In the following content, we use "knn", "wknn", and "awknn" to represent the methods of (4), (6), and Algorithm 4, respectively. In this experiment, the PC is equipped with CPU: Intel®Core™ i5-12400, RAM:32GB DDR5, and OS: Windows 11 Pro, and the software is equipped with PHP (version 8.2.4) and MariaDB (version 10.4.28). We conducted the experiments using knn, wknn, and awknn classifications.

Accuracy and precision are both types of measurement. KNN classification belongs to the accuracy type of measurement. The accuracy type is typically measured by true positive, true negative, false positive, or false negative values and extended into various metrics such as specificity, sensitivity, balanced accuracy, and negative predictive values. The measurement for KNN classification includes only true positive and false negative, representing the instances where the actual classification of the test data point matches or differs from the classification assigned by the classifier. These two measurements alone cannot calculate specificity, sensitivity, balanced accuracy, or negative predictive values. In the experimental results, the accuracy is as (7), where TP is the true positive value and FN is the false negative value.

$$\text{Accuracy} = \frac{\text{TP}}{\text{TP} + \text{FN}} \tag{7}$$

First, we show the classification time. Tables 2 and 3 provide the classification time of knn, wknn, and awknn in Type-1 and Type-2 based on different *size* values. In the Type-1 experiment, when *size* value is 2 to 128, to classify a test data point, the time required by knn is $5.24 \times 10^{-6}$ to $9.43 \times 10^{-3}$ s; the time required by wknn is $6.66 \times 10^{-6}$ to $9.90 \times 10^{-3}$ s; awknn takes $3.08 \times 10^{-6}$ to $1.79 \times 10^{-4}$ s. Similarly, in the Type-2 experiment, the time required by knn is $3.17 \times 10^{-3}$ to $2.29 \times 10^{-1}$ s; the time required by wknn is $3.21 \times 10^{-3}$ to $2.31 \times 10^{-1}$ s; the time required by awknn is $5.51 \times 10^{-6}$ to $4.07 \times 10^{-2}$ s.

**Table 2.** Classification time based on different sizes in Type-1, where *size* is the average number of training data points per class.

| Classification \ *Size* | 2 | 4 | 8 | 16 | 32 | 64 | 128 |
|---|---|---|---|---|---|---|---|
| knn | $5.24 \times 10^{-6}$ | $1.42 \times 10^{-5}$ | $4.36 \times 10^{-5}$ | $1.52 \times 10^{-4}$ | $5.80 \times 10^{-4}$ | $2.30 \times 10^{-3}$ | $9.43 \times 10^{-3}$ |
| wknn | $6.66 \times 10^{-6}$ | $1.57 \times 10^{-5}$ | $5.17 \times 10^{-5}$ | $1.61 \times 10^{-4}$ | $5.95 \times 10^{-4}$ | $2.41 \times 10^{-3}$ | $9.90 \times 10^{-3}$ |
| awknn | $3.08 \times 10^{-6}$ | $5.42 \times 10^{-6}$ | $6.25 \times 10^{-6}$ | $1.27 \times 10^{-5}$ | $1.96 \times 10^{-5}$ | $5.07 \times 10^{-5}$ | $1.79 \times 10^{-4}$ |

Note: unit—seconds.Page: 13

**Table 3.** Classification time based on different size in Type-2.

| Classification \ *Size* | 2 | 4 | 8 | 16 | 32 | 64 | 128 |
|---|---|---|---|---|---|---|---|
| knn | $3.17 \times 10^{-3}$ | $1.34 \times 10^{-2}$ | $6.38 \times 10^{-2}$ | $2.74 \times 10^{-1}$ | $1.12 \times 10^{0}$ | $4.91 \times 10^{0}$ | $2.29 \times 10^{1}$ |
| wknn | $3.21 \times 10^{-3}$ | $1.37 \times 10^{-2}$ | $6.48 \times 10^{-2}$ | $2.79 \times 10^{-1}$ | $1.14 \times 10^{0}$ | $5.02 \times 10^{0}$ | $2.31 \times 10^{1}$ |
| awknn | $5.51 \times 10^{-5}$ | $2.47 \times 10^{-4}$ | $5.39 \times 10^{-4}$ | $2.09 \times 10^{-3}$ | $7.55 \times 10^{-3}$ | $2.96 \times 10^{-2}$ | $4.07 \times 10^{-2}$ |

Note: unit—seconds.Page: 13

Figure 5 provides comparisons of classification times for Type-1 and Type-2 between wknn versus knn and knn versus awknn. Figure 5a shows the result of wknn and knn. For classification time, in Type-1, wknn is 1.03 to 1.27 times larger than knn; in Type-2, knn is 1.01 to 1.02 times larger than awknn. Figure 5b shows the results of knn and awknn. In Type-1, knn is 2.16 to 52.68 times that of awknn; in Type-2, knn is 57.53 to 562.65 times that of awknn. The above results show that the classification time of wknn is slightly larger than that of knn, and the classification time of knn is much larger than that of awknn. As the training dataset increases, this difference becomes more considerable. The reason is as follows. When knn's technology is used for a test point, it will search each training data

point and calculate the Euclidean distance of the two points. Then, it sorts the training data points based on the Euclidean distance and obtains the closest $k$ neighbors. The technology of wknn is similar to the technology of knn, but it also needs to convert the value of Euclidean distance into a multiplicative inverse. This process causes the technology of wknn to be more complicated than the technology of knn, so the classification time of wknn is also slightly higher than that of knn. It is worth mentioning that both knn and wknn technologies require sorting all training points, which is the most time-consuming part of these two technologies. The search procedure of awknn does not need to calculate the Euclidean distance between the test point and the training point. Therefore, when comparing search procedures, the performance of awknn is better than that of wknn or knn. When comparing the ranking process, awknn only processes a few candidate points, while wknn or knn process all training data points. This is why the classification time of awknn is much lower than that of wknn and knn.

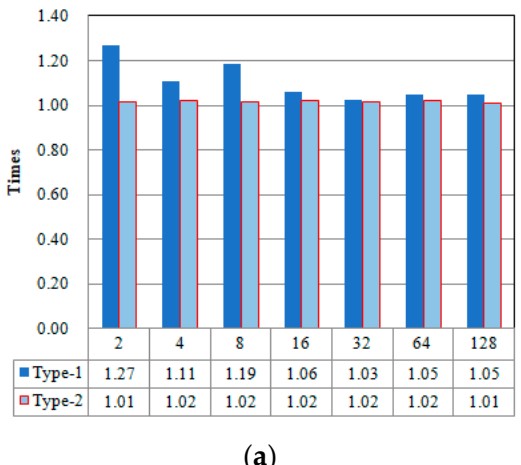

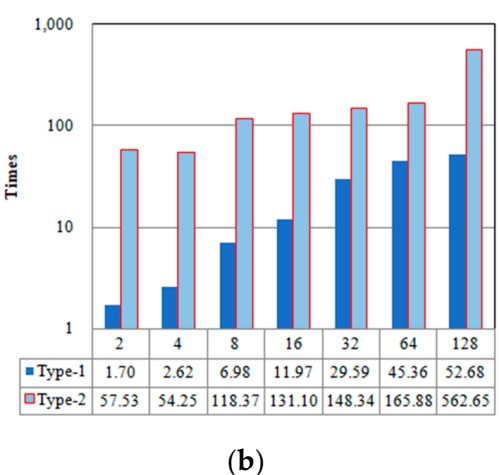

| | (a) | | (b) | |

**Figure 5.** Cross-comparison of classification times. (**a**) wknn and knn; (**b**) knn and awknn.

The accuracy of a classification method is generally higher with more training data points, but this comes at the cost of increased classification time. The choice of the different $k$ values also affects accuracy. Therefore, the critical observation for evaluating a classification method lies in understanding the relationship between the number of training data points and accuracy under different $k$ values. Hence, Figure 6 or Figure 7 illustrate the accuracy provided by different classification methods with varying $k$ values as the number of training data points increases in Type-1 or Type-2.

Generally, the classification accuracy provided by KNN technology will increase with the increase in training data points. In Figures 6 and 7, the classification accuracy provided by knn, wknn, and awknn conforms to this characteristic. For example, in Figure 6a, when *size* value is 2, the classification accuracy of knn is between 31.84% and 66.67%; the classification accuracy of wknn is between 54.05% and 66.67%; the classification accuracy of awknn is between 54.05% and 66.67%. The accuracy ranges from 53.96% to 66.52%. As the training data points (*size*) increase, the accuracy of these methods will also increase. In Figure 6g, when *size* value is 128, the classification accuracy of knn ranges from 94.17% to 95.92%; the classification accuracy of wknn ranges from 95.65% to 96.03%; the classification accuracy of awknn ranges from 95.65% to 96.04%.

Under the same classification conditions, the classification accuracy provided by KNN technology will decrease as the number of categories increases. The results of comparing Figures 6 and 7 are also consistent with this characteristic. Taking Figures 6c and 7c as an example, the accuracy of knn in Figure 6c ranges from 73.36% to 84.45%; the accuracy of knn in Figure 7c ranges from 68.56% to 77.02%.

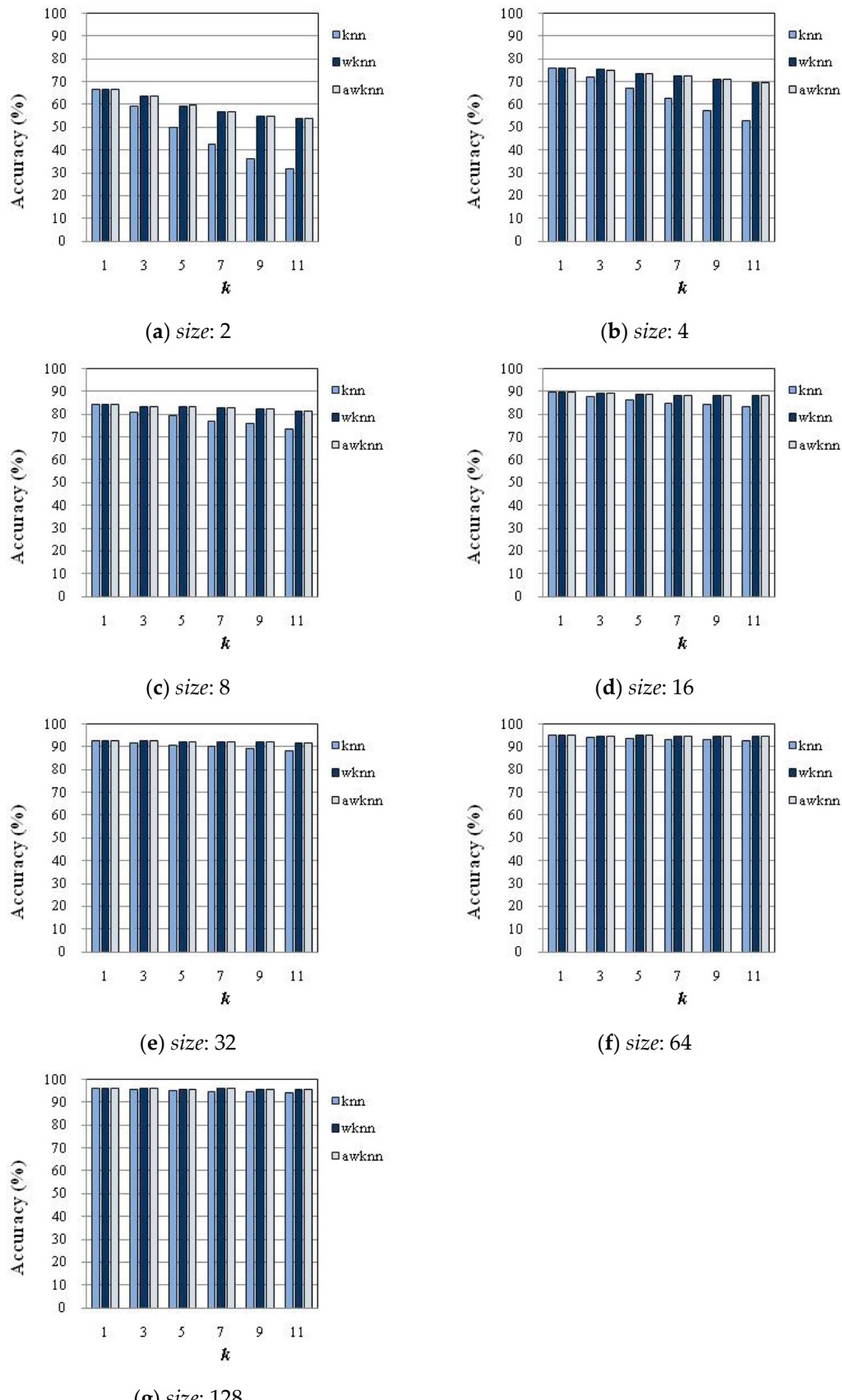

**Figure 6.** Classification accuracy based on Type-1 for different *size* values with different *k* values.

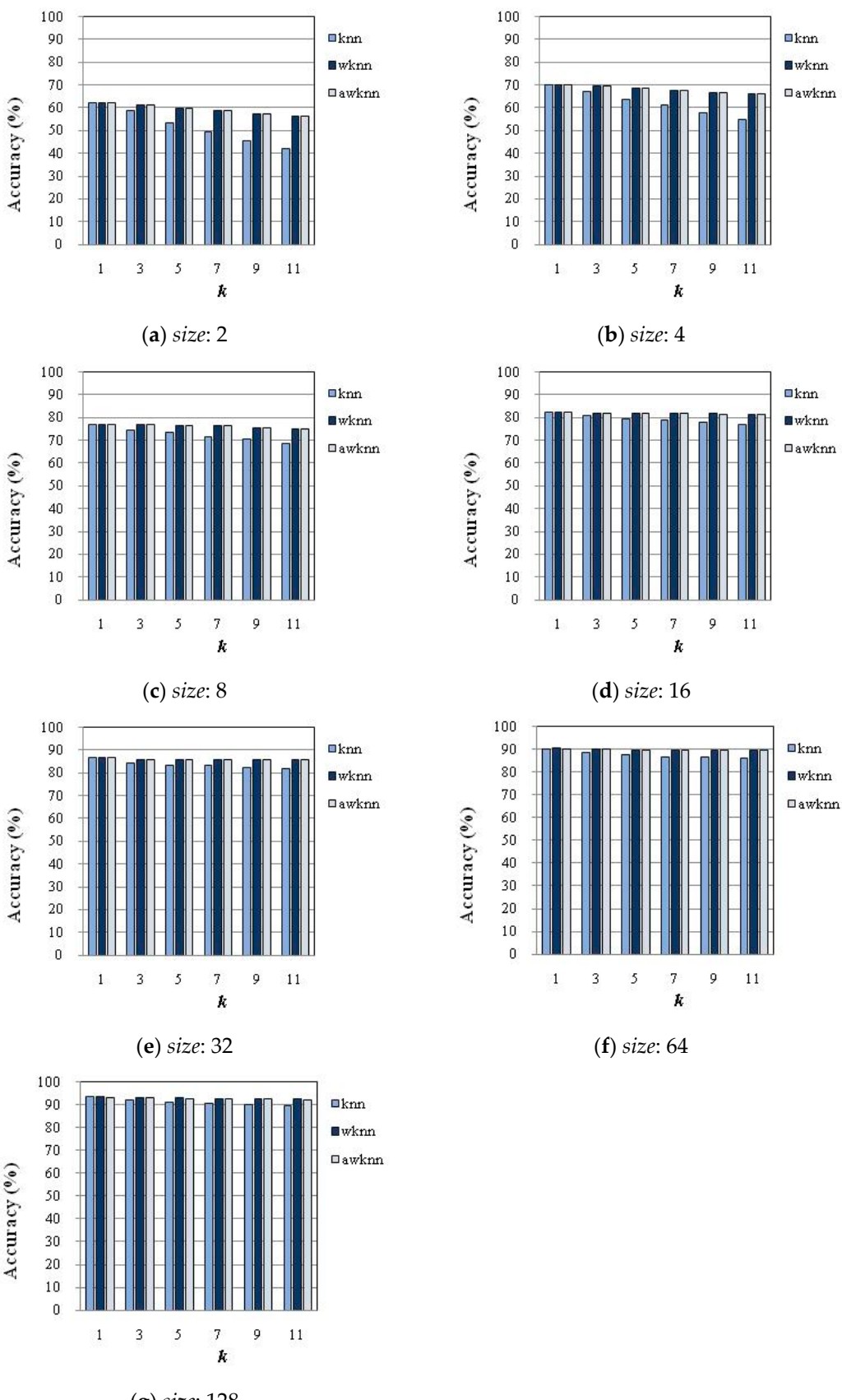

**Figure 7.** Classification accuracy based on Type-2 for different *size* values with different *k* values.

The results from Figures 6 and 7 also demonstrate that, even with the same number of training data points, different $k$ values can still impact accuracy. Consequently, Table 4 or Table 5 presents the average accuracy of each classification method under a fixed *size* value for Type-1 or Type-2, considering $k$ values ranging from 2 to 128. For example, in Figure 6a, under different $k$ values, the classification accuracy of knn ranges from 31.84% to 66.67%, with an average value of 47.79%.

**Table 4.** Average accuracy of different $k$ values in Type-1.

| Classification | *Size* 2 | 4 | 8 | 16 | 32 | 64 | 128 |
|---|---|---|---|---|---|---|---|
| knn | 47.79 | 64.75 | 78.54 | 86.07 | 90.57 | 93.70 | 95.00 |
| wknn | 59.31 (11.52) | 72.91 (8.16) | 82.98 (4.44) | 88.82 (2.75) | 92.27 (1.70) | 94.82 (1.12) | 95.85 (0.85) |
| wknn | 59.30 (11.51) | 72.86 (8.11) | 82.94 (4.40) | 88.79 (2.72) | 92.27 (1.70) | 94.83 (1.13) | 95.85 (0.85) |

Note: unit—%.

**Table 5.** Average accuracy of different $k$ values in Type-2.

| Classification | *Size* 2 | 4 | 8 | 16 | 32 | 64 | 128 |
|---|---|---|---|---|---|---|---|
| knn | 51.88 | 62.53 | 72.62 | 79.30 | 83.64 | 87.56 | 91.20 |
| wknn | 59.21 (7.33) | 68.11 (5.58) | 76.17 (3.55) | 81.85 (2.55) | 85.98 (2.34) | 89.77 (2.21) | 92.84 (1.64) |
| awknn | 59.20 (7.32) | 68.11 (5.58) | 76.17 (3.55) | 81.84 (2.54) | 85.98 (2.34) | 89.77 (2.21) | 92.77 (1.57) |

Note: unit—%.

The results from Figures 6 and 7 also demonstrate that, even with the same number of training data points, different $k$ values can still impact accuracy. Consequently, Tables 4 and 5 present the average accuracy of each classification method under a fixed *size* value for Type-1 or Type-2, considering $k$ values ranging from 2 to 128. Moreover, Tables 4 and 5 provide the average accuracy based on different *size* values in Type-1 and Type-2. Through the analysis of Tables 4 and 5, we can better understand the relationship between training data points and classification accuracy.

Tables 4 and 5 provide the average accuracies of different $k$ values in Type-1 and Type-2. The values in parentheses are the differences in accuracy between wknn (or awknn) and knn. KNN technology is used in many fields. In most fields, the technology of weighting KNN can improve classification accuracy, but it is not possible. For example, in Table 1, for datasets such as Yeast, Arcene, and CLLAUB, weighting KNN technology enlarges the classification complexity but is not reflected in the accuracy. Fortunately, the technology of weighting KNN achieves the expected results in this field.

Next, we discuss the classification accuracy of KNN or weighting KNN technology. From the results in Table 1, we know that this technology has limited performance in some fields. For example, in the Letter dataset, under 26 classes with 249 average training data points per class conditions, the accuracy is only 4.71% in KNN and 5.38% in weighting. In this field, despite limited training data points, such as *size* value of 2, wknn technology can achieve an accuracy of nearly 60%. In Table 1, German, with a *size* value of 69, and Ionosphere, with a *size* value of 23, can obtain an accuracy of 89.70% and 87.50%, respectively. It is worth mentioning that the number of classes in these datasets with better accuracy performance is 2. Type-1 has 12 classes, and the *size* value is 16; KNN technology can achieve similar accuracy. Even in Type-2, the number of classes is as high as 256, and when the dataset is sufficient, the accuracy can be as high as 90% or more. Therefore, KNN technology is very suitable for this field. The technology of awknn retains the advantages of weighting KNN technology, and the classification time is also superior.

## 6. Conclusions

In this paper, we have planned a strategy, including positioning and classification phases, which can be used when epidemic management or other applications need to track the location of some targets or people. In positioning phases, the areas where the target is located can be obtained through the latitude and longitude coordinates of targets. This strategy can determine the required training data points according to the accuracy required by different applications. The classification phase can start when the training data points are sufficient. In this phase, weighting KNN technology is introduced into this field to calculate the area where the target is located. Compared with the classification accuracy of KNN in other fields, KNN or weighting KNN technology is very suitable for this application. The classification time of KNN-based technology is $O(n_{TD}^2)$, where $n_{TD}$ is training data points. For this phase, we have planned an adaptive KNN algorithm. The experimental results show no significant difference between the accuracy provided by this algorithm and the accuracy provided by weighting KNN. When a target object needs to determine its area, this algorithm will plan a square range with the target as the center and, through a search of the training data points, obtain the data points within this range as candidate points when voting in KNN. When the candidate points are insufficient, this algorithm will adaptively adjust the square range to meet the needs. Generally, KNN technology will treat all training data points as voting candidate points. However, in the algorithm we planned, only a few training data points will become candidate points, which achieves $O(n_{TD})$ classification time. When the epidemic spreads, many people must be classified by area. The traditional KNN technology may not be able to handle the classification time, but our method can meet the needs. At present, although we have good performance in classification accuracy and classification time, in the future, we still aim to improve accuracy and reduce classification time.

Emerging infectious diseases have been an essential topic in epidemiology in recent years [49,50]. Preventing and slowing down the spread of these infectious diseases is crucial at the onset of an epidemic. Once the situation reaches a full-blown outbreak, it consumes significant medical resources and adversely affects public health. In public health, conducting systematic epidemic investigations, accurately identifying the footprint of virus spread, and deducing models for the spread of the epidemic are crucial but challenging tasks. The challenges arise from the unknown transmission pathways and transmission capabilities of emerging infectious diseases. Only when microbiologists discover the characteristics of these microorganisms will we rely on more traditional methods, such as contagious disease reporting and telephone-based contact tracing, which are less efficient, resource-intensive, and prone to oversight. This approach can create vulnerabilities in epidemic control, leading to widespread dissemination. To address these issues, in addition to the traditional roles played by healthcare professionals, public health experts, and government agencies in epidemic prevention, the development of more efficient epidemic monitoring and predictive diffusion model analysis using technology and the internet has become increasingly important, which is also a project valued and emphasized by us. We hope this research can help epidemic management understand the spread of these pathogens and enable us to make predictions and preparations earlier, significantly as the infection numbers rapidly increase.

**Author Contributions:** Conceptualization, J.-S.C.; software, C.-M.K.; validation, J.-S.C.; formal analysis, J.-S.C.; investigation, J.-S.C.; data curation, C.-M.K.; writing—original draft, J.-S.C. All authors have read and agreed to the published version of the manuscript.

**Funding:** This research received no external funding.

**Data Availability Statement:** Data are contained within the article.

**Conflicts of Interest:** The authors declare no conflict of interest.

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
