# Peer review of "An Efficient GNSS Coordinate Classification Strategy with an Adaptive KNN Algorithm for Epidemic Management"

_mathematics, doi:10.3390/math12040536_

Round 1
Reviewer 1 Report
Comments and Suggestions for Authors
In the article, the authors propose a system model for classifying geographical points. It consists in two stages: positioning and classifying. Positioning stage implements in point-in-polygon algorithm, classifying stage implements in adaptive weighted kNN algorithm. The manuscript is well structured and illustrated, the proportion of self-citations in the reference list is acceptable.
The main issue is that the study lacks scientific novelty. It seems that the research consists of using well-known methods. Authors should make their contribution to the study more clear. References should be made to the state-of-the-art methods on both stages (such as weighted kNN, for instance) and the improvements made to them should be clearly indicated. Please, review existing adaptive kNN methods and highlight how your method differs from them.
Other comments:
Line 141 – “the methods in [P1-P6] may fall short” – wrong references
Line 153 – “This study in [P9]…” – wrong reference
Line 162 – “time complexity of O(n TD ), where n is the dataset size.” N is described instead of nTD. And this does not coincide with the designation in the Conclusion section.
It would be nice to show the dimensions of the datasets in Table 1.
In Algorithm 1, result := 1 is repeated twice (lines 8 and 9).
In (4), the g’, gax, gay, gbx, gby are not denoted and a reader must guess.
Line 354 – “…we use knn, wknn, and awknn to represent the KNN classification”. Please, provide the corresponding references.
Please, format the reference list according to the journal requirements.
Comments on the Quality of English LanguageThere are many typos and inaccuracies in the text. For instance:
Line 41 – word repetition “altitude (altitude)”
Line 54 – “Calculate whether a point in the polygon Interior is a PIP problem in geometry.”
Line 151 – “…it considers the computer's computing power…”
Line 372 – “In Type-1, wknn is 1.03 to 1.27 times that of knn.”
Line 402 – “In Figure 6(g), when the average size is 128, the classification of knn The accuracy ranges from 94.17% to 95.92%”
Author Response
Please see the attachment.
I try my best. Please forgive me for any imperfections.

Reviewer 2 Report
Comments and Suggestions for Authors
The review is attached

Round 2
Reviewer 1 Report
Comments and Suggestions for Authors
The authors made many corrections, but not all the necessary ones. Please, review existing adaptive kNN methods and highlight how your method differs from them.
Line 104 “Additionally, we incorporate the idea of weighting in KNN into this study”. This phrase can be confusing because the reader may get the impression that the authors are introducing the idea of a weighted kNN. To avoid this, it is necessary to provide a reference to well-known weighted kNN methods (for example S. Dudani, “The distance-weighted k-nearest-neighbor rule” (1976) or others) and to make an accent on the differences between the author’s method and the well-known ones.
Reviewer 2 Report
Comments and Suggestions for Authors
The authors consider all of my recommendation and comments.
I have no other recommendation
Author Response
Thanks for your suggestions.
Round 3
Reviewer 1 Report
Comments and Suggestions for Authors
The article can be published